# The Prevalence of β-Thalassemia and Other Hemoglobinopathies in Kuwaiti Premarital Screening Program: An 11-Year Experience

**DOI:** 10.3390/jpm11100980

**Published:** 2021-09-29

**Authors:** Najat Rouh AlDeen, Asmaa A Osman, Monira J Alhabashi, Rasha Al Khaldi, Hassan Alawadi, Maha K Alromh, Eiman G Alyafai, Nagihan Akbulut-Jeradi

**Affiliations:** 1Hematology Department, Maternity Hospital, P.O. Box 4078, Sabah 13001, Kuwait; dr.naar90@gmail.com (N.R.A.); Asmaa_sayed75@yahoo.com (A.A.O.); Saja18@hotmail.com (M.J.A.); 2ATC Learn Center, Advanced Technology Company, P.O. Box 44558, Hawalli 32060, Kuwait; rasha.m@atc.com.kw; 3Social Health Administration, Kuwait Premarital Center, P.O. Box 4078, Sabah 13001, Kuwait; ptc@moh.gov.kw (H.A.); mkhalromh@gmail.com (M.K.A.); Q8tpmc@gmail.com (E.G.A.)

**Keywords:** premarital screening (PMS), β-thalassemia, Sickle cell, other hemoglobinopathies, Kuwait

## Abstract

This study aims to estimate the prevalence rates of β-thalassemia and Sickle cell disorders in the adult population screened (*n* = 275,819) as part of the Kuwaiti National Premarital Screening Program. All the individuals who applied for a marriage license during the years 2009 and 2020 were covered by the program. A network of four reception centers in the Ministry of Health facilities and one Premarital Diagnostic Laboratory (PDL) in Maternity Hospital were involved in performing all investigations for hemoglobinopathies. The total number of individuals identified with β-thal trait was 5861 (2.12%), while 22 individuals (0.008%) were diagnosed with β-thal disease. A total of 5003 subjects (1.81%) were carrying the Sickle cell trait, while 172 subjects (0.062%) had Sickle cell disease including Sickle cell anemia (SCA). Results showed that the program succeeded indeed in preventing the marriage of 50.4% of risky couples by issuing unsafe marriage certificates. Yet more efforts are needed to improve the program’s main objective of decreasing high-risk marriages. In particular, health care systems should be ameliorated in a way to intensify the counselling mechanism for the high-risk couples, strengthen the awareness of the general population and induce earlier age screening policies.

## 1. Introduction

Hemoglobinopathies are one of the most common monogenic inherited blood disorders affecting the structure, function or production of hemoglobin molecule. Hemoglobin disorders such as β-thalassemia and Sickle cell disease (SCD) affect millions of individuals worldwide. They are characterized by severe clinical heterogeneity, complicating patients’ management and treatment. β-thalassemias are clinically often associated with profound anemia, jaundice, expanded bone marrow volume, siderosis, cardiomegaly and splenomegaly. A thalassemia patient requires multiple repeated blood transfusions and hence subjected to peroxidative tissue injury due to secondary iron overload [1]. Sickle cell disease (SCD) is a group of genetic disorders in which hemoglobin is structurally abnormal, resulting in the episodic formation of sickle-shaped red blood cells (RBCs) and a wide range of clinical manifestations. β-thalassemias and Sickle cell disease are now understood to be disorders of global importance causing significant morbidity and mortality, as well as affecting the economic and healthcare status negatively. In light of this, premarital screening (PMS) has been implemented as one of the major strategies for prevention of genetic disorders and infectious diseases [2,3]. PMS, supported by genetic counseling as well as education, can substantially prevent hemoglobinopathy disorders [4,5] by detecting the carriers of hemoglobinopathies that lead to affected offspring of couples that are planning to marry.

Preventive screening programs such as PMS have been implemented in several countries worldwide, and aim to identify carriers of the hemoglobin disorders in the population to assess the risk of having children with a severe form of disease and reduce the prevalence of β-thal and SCD [6]. Countries in the Middle East have established premarital screening and genetic counseling (PMSGC) programs as a mandatory step prior to receiving the marriage license and offer genetic counseling to at-risk couples for hemoglobinopathy disorders [7].

In Kuwait, a mandatory premarital screening program was initiated in 2009 and involves screening individuals intending to get married for β-thal, SCD, Human Immunodeficiency Virus (HIV), Syphilis and Hepatitis B and C. In this study, we present the findings of eleven years of PMS in Kuwait regarding β-thal and SCD, resulting in an estimation of the prevalence of hemoglobinopathies in the entire adult population as well as the effect of the program among high-risk couples. The prevalence rates for β-thal and SCD trait among premarital couples was found to be 2.12% and 1.81%, respectively, indicating a great need to reduce the burden of these hemoglobinopathies in Kuwait by successful and efficient premarital screening program.

## 2. Materials and Methods

### 2.1. Premarital Screening Program

In September 2009, H.H. the former Emir (Sabah Al-Ahmad Al-Jabeer Al-Sabah) passed the decree of the Premarital Screening Law No.31/2008 which involves mandatory screening of Kuwaiti individuals intending to get married. This article reports the impact of eleven years of premarital screening on the prevalence of hemoglobinopathies in Kuwait. The screening program offers free testing/counseling for local couples applying for a marriage license in Kuwait and aims to reduce the number of marriages between couples who are found to be affected or carriers of the aforementioned diseases.

The first step of the PMS program was the referral by the Legal Authentications Administration (Ministry of Justice). Prospective couples were directed to one of the four Ministry of Health (MOH) designated Premarital Reception Centers: Sabah area (Main Center), Ahmadi PMS (Mubarak Al-Kabeer Health Center), Farwaniya PMS (Menahi Al-Osaimi Health Center) and Jahra PMS (Al-Jahra Health Center). Demographic information (age, place of residence, place of birth and contact numbers, medical history) was collected by these centers which were equipped with medical personnel and supplies. Blood samples were collected in tubes containing EDTA anticoagulant and plain tube without additives/anticoagulant to be sent to the Premarital Diagnostic Laboratory (PDL) in Maternity Hospital, which is the only center authorized by MOH to perform all investigations for hemoglobinopathies and syphilis. The screening protocol (screening cascade is shown in Figure 1) included complete blood counts (CBC) (DxH 600, Beckman Coulter, Brea, CA, USA), absolute reticulocyte count, peripheral blood film, high-performance liquid chromatography (HPLC) (Bio-Rad Variant II, Hercules, CA, USA) as well as a sickling test using sodium dithionite and serum ferritin. The CBC and HPLC tests were compulsory for all samples received by PDL. HPLC was used as the main diagnostic method to reveal hemoglobinopathy disorders such as β-thal, SCD and different Hb variants. Diagnosis of β-thal trait was considered if a person had a mean corpuscular volume (MCV) of ≤80 fL, a mean corpuscular hemoglobin (MCH) of ≤27 pg and a hemoglobin A2 level of ≥3.5%. Cases presented with normal CBC parameters and normal serum ferritin and HbA2 levels between 3.3% and 3.4% were considered as borderline cases (possible silent β-thalassemia) needing family and genotyping studies. For diagnosis of the Sickle cell trait, it was required to show the presence of HbS with positive manual sickling. The screening of the hemoglobinopathy program was further supported by confirmatory assays, namely hemoglobin (Hb) gel electrophoresis (Interlab, Roma, Italy) and capillary electrophoresis (Sebia, Lisses, France) since 2014 and 2016 respectively. Starting in 2014, selected samples were sent to the Maternity Hospital Molecular Genetics Laboratory for further in-depth investigation; their results will be reported separately.

All the couples found to have a low risk were issued compatibility certificates (“Safe marriage”). On the other hand, when both partners were found to be positive for Sickle cell trait/disease and/or thalassemia trait/disease and/or other hemoglobin variant trait/disease they were designated as high risk couples and referred to a genetic counseling clinic. There, the counselor would explain the situation to both partners and issue a certificate of incompatibility (“Unsafe marriage”). Although the primary aim of counseling was to avoid high-risk marriages, the couples were free to marry regardless of their test results as long as the female was above 21 years old age.

### 2.2. Ethical Consideration

The study was approved by the MOH ethical committee with the code of 2019/1151. The privacy and rights of the individuals were protected by appropriate protocols; no patient identifiers were used in any part of the study.

### 2.3. Data Management and Statistical Analysis

The dataset was thoroughly examined for integrity, inaccuracies, inconsistencies and invalid entries. Necessary changes were made to correct data errors and recode variables as needed. Subjects’ data were recorded in a questionnaire and analyzed using SPSS v25 for Windows (IBM, Chicago, IL, USA). Continuous variables were summarized with descriptive statistics (N, mean, standard deviation, range). Categorical variables were summarized with frequency counts and percentages within each category, or between levels of a category as appropriate. The prevalence rates were calculated separately for disease and carrier statues within each condition and estimated as the number of affected individuals divided by the number of tested individuals per 1000 population. A *p* value ˂ 0.05 was accepted as statistically significant.

## 3. Results

### 3.1. Population Characteristics

During the 11-year reporting period (August 2009–August 2020), a total of 275,819 individuals were screened within the program, with the majority (58.96%) of program attendees in the 25–34 year-age group. Almost 90% of the individuals were holding Kuwaiti nationality (KN). The female:male ratio was 51.0%:49.0% and 76.32% of the candidates applied to the premarital program for their first marriage. Most of the participants (53.06%) had a university degree or above. Table 1 summarizes the sociodemographic characteristics of the participants.

The overall study population of 275,819 individuals approached the four different premarital screening centers. The highest proportion of screened KN individuals was from the Ahmadi region (35.62%), followed by Sabah (25.09%), Farwaniya (17.51%) and Jahra (11.60%) (Figure 2). On the other hand, the rate of screened Non-Kuwaiti nationality (NKN) individuals was about an average of 3.41% in Sabah region, 2.69% in Ahmadi, 2.36% in Jahra and 1.71% in Farwaniya.

### 3.2. Prevalence Rate, Time Trends and Distribution of β-Thalassemia and Sickle Cell Hemoglobinopathies

The total number of individuals identified with β-thal trait was 5861 (2.12%), while 22 individuals (0.008%) were identified with β-thal disease. β-thalassemia prevalence varied by region; 17.97% and 16.84% of β-thalassemia trait cases were from Sabah and Ahmadi regions, respectively, followed by Farwaniya (8.12%) and Jahra (5.65%) regions. Regarding Sickle cell hemoglobinopathies, 5003 subjects (1.81%) were carrying the Sickle cell trait, while 172 subjects (0.062%) had Sickle cell disease including Sickle cell anemia (SCA). The rates of Sickle cell trait were highest in the Sabah region (16.45%) followed by Ahmadi (13.52%), Farwaniya (6.22%) and Jahra (1.04%) regions (Figure 3).

In the 11-year program period, the average prevalence rate per 1000 for β-thal and Sickle cell trait was 21.1 and 18.1, respectively. As shown in Table 2, the β-thal trait prevalence showed a fluctuating pattern with a minimum of 17.5 (95% CI = 15.9–19.2) observed in the year 2010. This is most probably related to the improvement in diagnostic tools and an increasing suspicion of silent β-thal trait. On the other hand, the prevalence rate for Sickle cell trait showed a clear decreasing trend, from 21.0 in 2009 to 14.1 in August 2020, with a minimum of 13.5 (95% CI = 12.1–14.9) per 1000 in 2019.

### 3.3. Other Hemoglobin Abnormalities, Borderline HbA2 and Iron Deficiency Anemia Cases

Following β-thal and Sickle cell traits, 747 cases of Hb D trait (0.27%), 145 cases of Hb E trait (0.053%) and 49 cases of Hb C trait (0.018%) were the most prevalent ones. The overall prevalence rate of Hb H disease during the 11-year period was 70 cases (0.025%), which are identified by the presence of Hb H inclusion and Hb H peak in HPLC that most likely represent a deletional type of Hb H disease followed by Hb D disease (0.010%) and Hb C disease (0.001%). Almost 0.1% of the screened samples showed an unknown peak in HPLC analysis, or a band with altered mobility in alkaline electrophoresis analysis; some of these samples were referred to the Molecular Genetics Unit.

All abnormal cases of hemoglobin variants recorded during the screening program in the different PMS centers are shown in Table 3. A total of 10% of the samples gave HbA2 values between 3.3% to 3.4% and were considered borderline cases; in such cases, especially when a partner HPLC result reported any abnormal band, the couples were referred to the Molecular Genetics Unit, in order to avoid missing an “at risk” couple. The implementation of genotyping to screen hemoglobinopathies in 2014 necessitated readjustment of the borderline HbA2 cut-off between 3.2% to 3.5% within the scope of the program. The results generated by the Molecular Genetics Unit will be represented in another paper.

Our program also involved screening all the samples with low MCV and MCH values to be tested for serum ferritin; 7.86% of the total screened individuals were diagnosed with iron deficiency anemia. Of these, 94.6% were females; a gender distribution of the total cases is shown in Figure 4.

### 3.4. Prevalence of Safe and Unsafe Marriage

Of all couples, 137,744 (98%) were issued a safe marriage certificate, 1485 (1%) were issued an unsafe marriage certificate and 1064 (1%) were not issued a marriage certificate and their marriage was stopped; of these who were issued unsafe certificates, 358 (24%) were due to hemoglobinopathies in both partners.

The other group with no marriage certificate (1064 couples) can be divided into three groups. First, for 387 cases (36%) the female was under 21 years of age; out of those, 144 were diagnosed with a blood disease. Second, for 510 cases (48%) there was a disagreement between couples for unsafe marriage; of these, 168 disagreed to proceed due to the risk of hemoglobinopathy. The remaining third group, 167 cases (16%) involved other non-hematological causes. Overall, it was noticed that there is an increasing trend towards disagreement between couples over the 11 years of the screening program (Table 4).

In order to study the effect of Kuwait Premarital Screening (KPMS) program on the future prevalence of hemoglobinopathies and whether it has a preventive and discouraging effect, from the available data the total unsafe results due to hemoglobinopathies in both parties in couples during the last 11 years was found to be around 670 couples with unsafe results. This included the following group: A total of 358 couples issued unsafe certificates, of which 332 couples (49.6%) proceeded to marriage despite the risk and 26 (3.8%) couples decided to stop the marriage; data supplied by the Ministry of Justice. A second group included 312 couples (46.5%) who stopped their marriage because of hemoglobinopathies either by law (144 couple) or disagreement (168 couple). Thus, the KPMS program succeeded to prevent and discourage around 50.4% of those couples with unsafe hemoglobinopathies results to proceed with the marriage, as shown in Table 4.

## 4. Discussion

Here we present the first published data from Kuwait Premarital Screening and Genetic Counseling Program (PMSGC) of hemoglobinopathies, reporting the prevalence and distribution of β-thal, SCD and other hemoglobinopathies in Kuwait in the period August 2009–August 2020. One strength of the Kuwaiti PMSGC program is the fact that it was made mandatory. This claim is derived by the cultural behavior of the population when adapting to new rules and changes to the habits that have been practiced or passed on from one generation to the next. Marriage is a culturally sensitive subject that follows traditions and cultural/religious rules, thus unless a mandate on inserting science in the process is put in place the society would have never adhered to the new rules on pre-testing. Additionally, the program offers free pre-testing for all local individuals (both partners are screened even if only one is Kuwaiti) seeking to get married. The program has a wide coverage involving around 2% of the total population every year, and provides valuable data that contributes to a more accurate assessment of the prevalence of hemoglobinopathy disorders in Kuwait. The results reveal a prevalence of β-thalassemia (2.12%) and Sickle cell trait (1.81%), comparable to those reported in a study conducted in the Gulf Cooperation Council [8,9]. On the other hand, the Kuwaiti prevalence is lower than those found in a study concerning Saudi Arabia (4.2% and 3.2%, respectively, among 488,315 individuals) [3]. Regarding β-thal, the Kuwaiti prevalence is much lower than those found in studies concerning Cyprus and Greece (16.4% and 7.4%, respectively) [10,11].

One of the important observations of this study is that 22 cases of β-thal major applied for PMS testing, which reflects a good standard of management of the β-thal major program in Kuwait for pediatric and early adulthood. On the other hand, it is worth mentioning that those 22 cases probably are not reflecting the actual number of adult patients with β-thal major in Kuwait.

The Kuwait PMSGC program was successful in preventing and discouraging at-risk marriages which will contribute to reducing the prevalence of affected offspring, especially with the newly implemented program of preimplantation genetic diagnosis (PGD) to be offered for free for high-risk marriages that received an un-safe marriage certificate and planned to continue proceed with their marriage. One of the primary goals of the Kuwait PMSGC program is to identify couples at risk of having a hemoglobinopathy-affected child and to reduce or even eliminate the presence of β-thal and SCD in newborns. However, achieving this goal is complicated and dependent on the presence of other preventive programs such as national prenatal screening and abortion, the implementation of which is restricted by ethical, religious, and societal values [12,13]. Evaluation of the effectiveness of the premarital screening program can benefit greatly from determining β-thal and SCD rates in newborns; therefore, a national neonatal screening program (which has not been implemented in Kuwait yet) and its integration with premarital screening is important to monitor the performance of the program, which will reflect of on the future incidence of hemoglobinopathies in Kuwaiti society. Besides screening premarital couples and newborns, an additional helpful strategy would be to offer a screening to young adults at an earlier stage (e.g., immediately after high school), so that they can discuss the issue in the early phase of the marriage proposal.

In addition to screening programs, health education is of great importance to achieve the set goals and should be focused on younger age groups. Studies among university students showed that knowledge of the PMS program, even though they were aware of the program, was low [14,15]. Organizing school lectures about hemoglobinopathy screening programs can improve the attitudes and understanding towards genetic counseling. For example, in Canada, 90–95% success was achieved by educating high school students on voluntary screening and genetic counselling over a 20-year period [16]. Thus, to improve the success of the program, effective genetic counseling and psychological support should be provided before marriage, to ensure that at-risk couples have a comprehensive awareness of the risks before the marriage decision has been made. To further increase productivity of the program and to collect the proper data to evaluate its success, a regular follow-up of high-risk couples needs to be established in the system. Therefore, cooperation with the Justice Department responsible for registering all marriages and issuing marriage certificates for all the marriages in Kuwait is necessary.

On the other hand, we found many factors play a major role in limiting the success of PMSGC in decreasing marriage among at-risk couples. One of the important risk factors affecting the occurrence of β-thal and SCD disorders is consanguineous marriages, in such cases the recessive genes will be able to survive and concentrate and retain the related diseases for generations. Consanguineous marriages have been described as the major reason for the higher prevalence of autosomal recessive disorders in populations of the Middle East and North Africa (MENA) region [17]. Indeed, Shawky et al. [18] showed that consanguineous parents have a higher chance of neonatal, post-neonatal and child mortality than those of non-relative marriage parents. However, in Kuwait it has been found that the consanguineous marriage is around 13.4%. Another important factor is the emotional and family pressure on premarital couples. We believe emotional and family pressure played a major role in their decision. Thus, although the program was successful in screening premarital couples, identifying those at risk of having children with Sickle cell trait/disease or thalassemia, and providing them with health education, positive (diminishing) effects on the incidence of hemoglobinopathy-related diseases and mortality are severely limited by not being able to prevent high-risk marriages because of these factors. Considering the fertility pattern of the population and the available health care system, practically all high-risk marriages are likely to produce children at high risk of hemoglobinopathies. An earlier study among 129 at-risk participants identified in premarital screening reported that only 2% cancelled their marriage proposal; in almost half of cases, cultural pressure was the main reason to proceed with marriage [19]. In contrast, a screening program for β-thalassemia has resulted in a decrease of birth prevalence by more than 95% in Cyprus [20], and similar success stories were found for Sardinia [21] and Turkey [22]. In other countries, such as Iran, Bahrain and Saudi Arabia, premarital screening programs have become widely implemented and some of these programs aim for prevention [7]. However, these programs have not always led to a reduction of affected births, due to the couples who marry despite being diagnosed as carriers, and due to a lack of prenatal diagnosis programs [7,23].

Together, the results show that education in the hemoglobinopathy prevention program and genetic counseling are still to be evaluated in detail. Knowledge and awareness of the PMSGC can be improved through relevant educational programs in schools and universities, as well as by governmental campaigns, especially on social media and the internet—the most efficient communication platforms nowadays. For example, using educational videos employing thalassemia patients might increase the awareness of the disease. With the collaboration of the government and non-government organizations, leaders in the community, religious leaders, parent organizations at schools and local health personnel, the program will develop community awareness of hemoglobinopathy diseases, which will increase its efficacy. The high morbidity related to hemoglobinopathies, chronic transfusion requirements, the high cost of chelating, organ damage, painful crisis, as well as the limited availability of stem cell therapies underline the importance of premarital screening programs.

Population screening programs for hemoglobinopathies have been implemented in many countries [24,25,26] for around 30 years. Such programs require financial resources of the receiving health care departments and medical laboratories involved. The long-term sustainability of the program depends on the financial and cost-effectiveness analysis of the program and the financial sources of the country. The Kuwait PMSGC program is provided free of charge to the local couples. PMSGC program laboratories are fully integrated into the existing health care system by using existing laboratories, and by training existing staff through existing training facilities. Additionally, the program was developed by existing staff in the Ministry of Health.

Genetic methods are in constant development, and preimplantation genetic diagnosis (PGD), carried out on embryos, could be an efficient tool in screening programs. PGD allows (at-risk) couples to exclude affected embryos and use only the healthy ones, and in this way, help decrease the prevalence of genetic disorders and their associated diseases. With PGD, couples avoid the trauma of leaving their partner as well as the ethical dilemma of terminating an affected pregnancy (when allowed by country laws). In this light, it is important to develop public health campaigns to highlight the different options. The resulting awareness may also promote a change in the governing laws of some countries, allowing at-risk couples to have children as they now have a viable, safe, successful, and cost-effective option to have unaffected offspring. Several studies have shown that integration of prenatal diagnosis options, such as PGD in prevention programs for hemoglobinopathies, are well established and successfully decrease Sickle cell and β-thalassemia prevalence [27,28,29]. Kuwait MOH offered a free-of-charge PGD service for any couple with an unsafe certificate, and even for any couples with the first baby of hemoglobinopathies. Furthermore, MOH is sponsoring PGD services in private labs as well.

Studies on Thalassemia and Sickle cell diseases have been covered in the literature since decades. Despite the existing efficient diagnostic and medical treatment strategies, lessening the burden of the diseases will always critically depend on prevention. There are several important components to evaluate a screening program, such as feasibility, access, cost, and, most importantly, measuring the effectiveness on the outcome of the diseases. Since Kuwait has a relatively small population (around 1.5 million inhabitants) to be screened through a PMS program, we preferred to have a centralized laboratory in order to maintain accuracy, reliability and validity as good as possible. Our study represents around 20% of the population and has new implemented strategies specific to the Kuwaiti people. In particular, we wanted to investigate the effect of the marriage prevention law for females less than 21 years old who got involved in an unsafe marriage. Our current study reveals the areas of weakness in the Kuwaiti PMS program and offers guidelines to health care providers to improve the primary aim of counseling: to prevent diseases by avoiding high-risk marriages. We believe the results and conclusions from our study will help involved parties to set up effective disease management strategies.

In conclusion, this study provides an update on recent trends in the occurrence of β-thal and SCD over a 11-year period in Kuwait (2009–2020). The obtained data will set the stage for continued monitoring of hemoglobinopathy disorders and will facilitate future evaluation of the effectiveness of existing preventive programs. Moreover, it will enable decision makers to prioritize new programs and implement policy changes. In order to better determine the genetic consequences, the Kuwait PMSGC program will require a longer timeframe, and would benefit greatly from following couples that proceed to marriage as well as following their offspring.

## Figures and Tables

**Figure 1 jpm-11-00980-f001:**
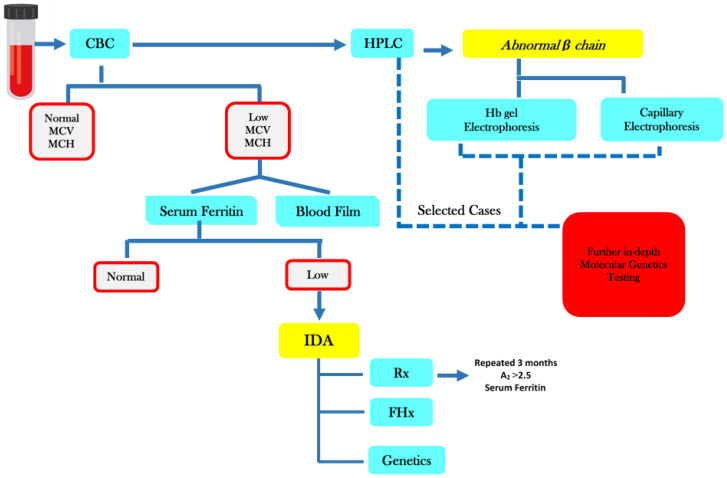
Diagnostic Cascade.

**Figure 2 jpm-11-00980-f002:**
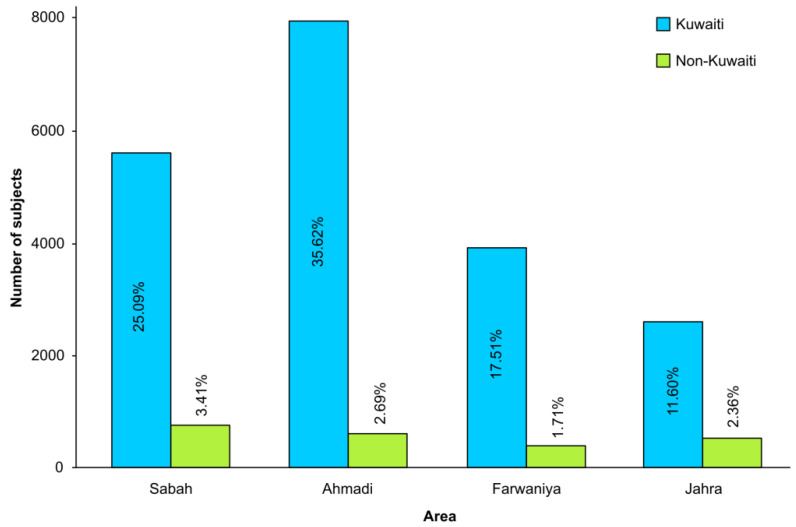
Percentages of screened Kuwaitis vs. Non-Kuwaitis according to the designated Premarital Reception Center.

**Figure 3 jpm-11-00980-f003:**
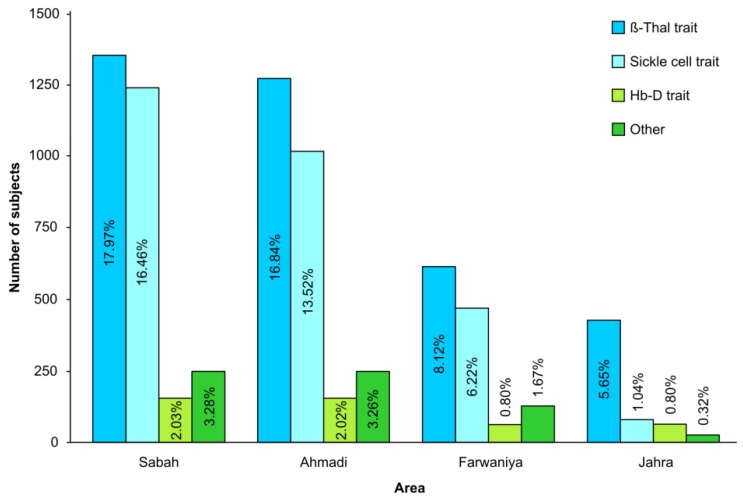
Abnormal cases detected by Hb electrophoresis according to the designated Premarital Reception Center from 2014 to August 2020.

**Figure 4 jpm-11-00980-f004:**
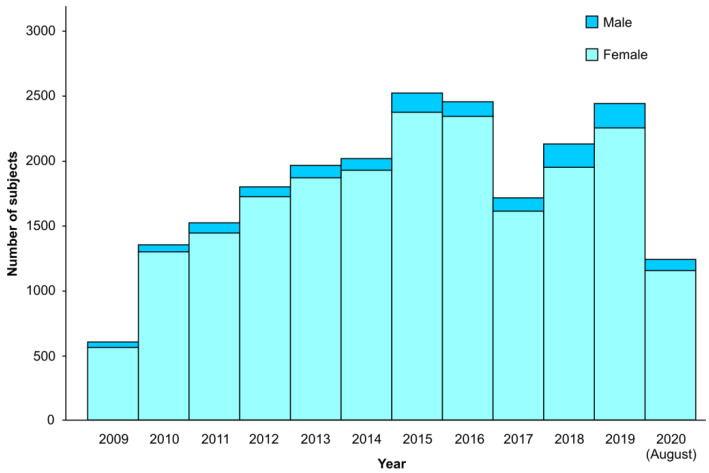
The total number of males and females with iron deficiency anemia cases from August 2009 to August 2020.

**Table 1 jpm-11-00980-t001:** Sociodemographic characteristics of the study population.

	Percentage (%)
Age Mean ± SD	
≤15 Years	0.02
15–24 Years	13.78
25–34 Years	58.96
≥35 Years	27.23
Nationality	
Kuwaiti (KN)	89.84
Non Kuwaiti (NKN)	10.16
Gender	
Female	51.03
Male	48.97
Level of Education	
Uneducated	0.06
Elementary	1.58
Intermediate	14.2
Secondary	31
Undergraduate and Post-Graduate	53.06
Marital Status	
Single	76.32
Married	4.06
Divorcee	19
Widow	0.63
Total Number = 275,819	

KN, Kuwaiti Nationality; NKN, Non-Kuwaiti Nationality; SD = Standard Deviation.

**Table 2 jpm-11-00980-t002:** The average prevalence rate per 1000 for β-thal and Sickle cell trait among the screened population.

		β-Thalassemia Trait			Sickle Cell Trait		
Year	Population Screened	Positive Cases	PR per 1000	95% CI PR per 1000	Positive Cases	PR per 1000	95% CI PR per 1000
2009	9763	190	19.46	16.9–22.0	205	21.00	18.0–24.0
2010	23,761	415	17.47	15.9–19.2	446	18.77	17.0–21.0
2011	25,050	488	19.48	17.8–21.3	445	17.76	16.0–19.0
2012	26,030	559	21.48	19.8–23.3	507	19.48	17.9–21.0
2013	26,823	554	20.65	19–22.4	531	19.8	18.0–20.0
2014	25,698	572	22.26	20.5–24.1	493	19.18	17.6–21.0
2015	25,693	556	21.64	19.9–23.0	504	19.62	17.9–21.4
2016	24,304	562	23.12	21.3–25.0	489	20.12	18.4–22.0
2017	24,047	645	26.82	24.8–28.9	485	20.17	18.5–22.0
2018	24,573	480	19.53	17.8–21.0	349	14.20	12.8–15.8
2019	25,780	557	21.61	19.9–23.0	348	13.50	12.1–14.9
2020-August	14,297	283	19.79	17.6–22.0	201	14.06	12.3–16.0
Total	275,819						

**Table 3 jpm-11-00980-t003:** Some of the hemoglobin variants detected during screening from 2009 to August 2020.

	Total Number of Cases	Percentage %
Hb D Trait	747	0.271
Hb D/D Disease	27	0.010
Hb D/B Disease	4	0.001
Hb D/C Disease	2	0.001
Hb S/D Disease	8	0.003
Hb S/E Disease	1	0
Hb S/C Disease	2	0.001
Hb E Trait	145	0.053
Hb E/E Disease	6	0.002
Hb C Trait	49	0.018
Hb C/C Disease	3	0.001
Hb H Disease	70	0.025
Hb H Disease + Sickle Cell Trait	4	0.001
HPFH	20	0.007
Hb DELTA/ B-Thal Trait *	22	0.008
Hb Lepore Trait	2	0.001
Hb Delta Thalassemia *	20	0.007
Hb C/B Thal Disease	1	0

Hb, Hemoglobin; HPFH, Hereditary persistence of fetal hemoglobin; * diagnosis by exclusion (as no molecular study available).

**Table 4 jpm-11-00980-t004:** Cumulative statistics for blood diseases for all PMC centers from 2010–2020.

Year		2010	2011	2012	2013	2014	2015	2016	2017	2018	2019	2020	Total	%	N.B
All Issued Certificates	Safe Marriage	11,632	12,319	12,856	13,315	12,747	12,752	12,039	11,937	12,209	12,782	13,156	137,744	98%	140,293% out of total issued and stop certificates
	Unsafe Marriage	186	129	159	132	124	143	141	118	108	121	124	1485	1%	
	Total	11,818	12,448	13,015	13,447	12,871	12,895	12,180	12,055	12,317	12,903	13,280	139,229	99%	
All Stop Certificates according to A reason	Age below 21 years	56	43	50	33	23	50	22	23	12	59	16	387	0.28%	
	Disagreement	47	46	61	46	55	23	40	50	43	16	83	510	0.4%	
	Referred for vaccination	X	27	34	17	20	18	12	9	12	11	7	167	0.33%	
	Total	103	116	145	96	98	91	74	82	67	86	106	1064	1%	
Unsafe Issue Certificates	Blood Diseases	18	23	36	30	25	39	41	32	36	41	37	358	0.3%	% out of total all issued certificates
Stop Certificates—Blood Diseases	Age < 21 years	12	14	22	10	10	14	16	18	11	12	5	144	13.5%	1064% out of total all stop certificates
	Disagree after counseling	4	11	13	10	12	17	15	17	18	23	28	168	15.8%	
	Total	16	25	35	20	22	31	31	35	29	35	33	312	29%	

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
