# Peer review of "The Prevalence of β-Thalassemia and Other Hemoglobinopathies in Kuwaiti Premarital Screening Program: An 11-Year Experience"

_jpm, 2021, doi:10.3390/jpm11100980_

Round 1
Reviewer 1 Report
The study of AlDeen et al. presents the rates of β-thalassemia and sickle cell disorders, in the adult population of Kuwait.
a) Please avoid plagiarism in the introduction: ''Premarital screening (PMS) is one of the major strategies for prevention of genetic disorders''
b) Enrich the genetic and epidemiological background of b-thalassemia and SCD, use the following citation for b-thalassemia: https://pubmed.ncbi.nlm.nih.gov/29068574/
c) For Ethical consideration is there a protocol number?
d) Improve figure 3 in terms of quality and presentation, i strongly believe can be better presented
e) Results section line 234: please rephrase: The results
generated by the Molecular Genetics Unit will be represented in another paper
f) yr : I know what means but PLEASE present the abbreviations
g) Discussion: How you claim that One strength of the Kuwaiti PMSGC program is the fact that it was made mandatory , please better explain and rephrase......
Author Response
Reviewer 1:
Comments and Suggestions for Authors
The study of AlDeen et al. presents the rates of β-thalassemia and sickle cell disorders, in the adult population of Kuwait.
a. Please avoid plagiarism in the introduction: ''Premarital screening (PMS) is one of the major strategies for prevention of genetic disorders''
Author’s response: We thank the reviewer for the valuable comments. The introduction has been amended.
**************
b. Enrich the genetic and epidemiological background of b-thalassemia and SCD, use the following citation for b-thalassemia: https://pubmed.ncbi.nlm.nih.gov/29068574/
Author’s response: We thank the reviewer for the comment. The genetic and epidemiological background have been enriched and the citation have been added upon recommendation.
**************
c. For Ethical consideration is there a protocol number?
Author’s response: We thank the reviewer for the comment. MOH Ethic Code approval has been submitted and inserted in the relevant section.
**************
d. Improve figure 3 in terms of quality and presentation, i strongly believe can be better presented
Author’s response: We thank the reviewer for the comment. Figure 3 as well as the other figures have been amended in terms of quality and presentation.
**************
e. Results section line 234: please rephrase: The results generated by the Molecular Genetics Unit will be represented in another paper
Author’s response: We thank the reviewer for the comment. The sentence has been rephrased as follows: Our data generated by the Molecular Genetics Unit are being analyzed and will be reported shortly.
**************
f. yr : I know what means but PLEASE present the abbreviations
Author’s response: We thank the reviewer for the comment. Abbreviations have been amended.
**************
g. Discussion: How you claim that One strength of the Kuwaiti PMSGC program is the fact that it was made mandatory, please better explain and rephrase.....
Author’s response: We thank the reviewer for the comment. The below is the amended rephrased section has been added to the manuscript. This claim is derived by the cultural behavior of the population when adapting to new rules and changes to the habits that have been practiced or passed on from generation to the next. Marriage is a culturally sensitive subject that follows traditions and cultural/religious rules and unless a mandate on inserting science in the process is put in place; the society would have never adhered to the new rules on pre-testing etc.
************************************************************
Reviewer 2 Report
AlDeen et al. present the results of eleven years of the Kuwaiti National Premarital Screening Program (2009-2020). A total of 275819 individuals intending to get married were screened for β-thalassemia (β-Thal), sickle cell disease (SCD), human immuno-deficiency virus (HIV), syphilis and hepatitis B and C. The authors report data about hemoglobinopathies (β-Thal and SCD) and the effect of the program among high-risk couples.
Finding reported provides an update on recent trends in the occurrence of β-thal (2.12%) and SCD (1.81%) in Kuwaiti, data are properly presented and the manuscript is well written but what is the novelty of the present paper? A multi-center experience could be interesting as long as it can produce new unexplored observations.
Being the originality of paper rather low, in my opinion it is not suitable for publication as an “original article” in the present form.
Author Response
Reviewer 2:
Comments and Suggestions for Authors
AlDeen et al. present the results of eleven years of the Kuwaiti National Premarital Screening Program (2009-2020). A total of 275819 individuals intending to get married were screened for β-thalassemia (β-Thal), sickle cell disease (SCD), human immuno-deficiency virus (HIV), syphilis and hepatitis B and C. The authors report data about hemoglobinopathies (β-Thal and SCD) and the effect of the program among high-risk couples.
Finding reported provides an update on recent trends in the occurrence of β-thal (2.12%) and SCD (1.81%) in Kuwaiti, data are properly presented and the manuscript is well written but what is the novelty of the present paper? A multi-center experience could be interesting as long as it can produce new unexplored observations.
Being the originality of paper rather low, in my opinion it is not suitable for publication as an “original article” in the present form.
Author’s response: We thank the reviewer for the comment. Studies on Thalassemia and Sickle cell diseases have been covered in the literature since decades. Despite the existing efficient diagnostic and medical treatment strategies, lessening the burden of the diseases will always criticially depend on prevention.
There are many premarital screening programs implemented in different countries with a similar approach; the published data has revealed progresses, obstacles and difficulties that each country faced during the implementation. This has helped other countries to establish/improve their current PMS proposals.
There are several important components to evaluate a screening program, such as feasibility, access, cost, and, most importantly, measuring the effectiveness on the outcome of the diseases. Since Kuwait has a relatively small population (around 1.5 million inhabitants) to be screened through a PMS program, we preferred to have a centralized laboratory in order to maintain accuracy, reliability and validity as good as possible. Our study represents around 20% of the population and has new implemented strategies specific to the Kuwaiti people. In particular, we wanted to investigate the effect of the marriage prevention law for females less than 21 years old who got involved in an unsafe marriage. Our current study reveals the areas of weakness in the current Kuwaiti PMS program and provides guidelines to health care providers to improve the primary aim of counseling: to prevent diseases by avoiding high-risk marriages.
Although the marriage prevention law is unique to Kuwait and has not been implemented in any of the Mediterranean or GCC Countries, we believe that the lessons learned from our study are important to the international community, also since it is one of the very few studies in the Gulf region and might be relevant to other countries in the Gulf region. Thus, in our opinion, it is very important to update the scientific and medical community on the national incidence rate of hemoglobinopathies, to reflect the quality of the prevention program as well as medical services. We believe the results and conclusions from our study will help involved parties to set up effective disease management strategies.
On the other hand with guidance of Reviewer 2, we added part of the explanation above to the conclusion.
Round 2
Reviewer 2 Report
In the light of the author's reply, the study takes on a different meaning.